∂ | **Open Peer Review** | Bacteriology | Research Article

# Multi-year comparison of VITEK MS performance for identification of rarely encountered pathogenic Gram-negative organisms (GNOs) in a large integrated Canadian healthcare region

D. L. Church,[1,2,3] T. Griener,[1,3] D. Gregson[1,2,3]

**ABSTRACT** This multi-year study (2014–2019) compared identification of rare and unusual Gram-negative organisms (GNOs) by matrix-assisted laser desorption ionization-time of flight mass spectrometry (MALDI-TOF MS) (VITEK MS, bioMérieux, Laval Que.) to 16S rRNA gene sequencing (16S) according to our laboratories routine workflow; 16S is done if initial MALDI-TOF MS gave discordant, wrong, or no results. GNB isolates were first analyzed by standard phenotypic methods and MALDI-TOF MS using direct deposit-full formic acid extraction; proteomics was repeated if no result occurred. Medically approved 16S analyses were done using fast protocols. Isolate sequences were analyzed using the Integrated Database Network System (IDNS3) bacterial database (SmartGene, Lausanne, Switzerland). Three hundred thirty-one GNOs including 251 (76%) aerobic Gram-negative bacilli (GNB), 63 (19%) fastidious Gram-negative coccobacilli (fGNCBs), and 17 (5%) *Campylobacterales* (CAMPB) isolates were recovered from 304 specimens; >1 isolate was recovered from 19 (6%). GNOs were mainly recovered from blood cultures (31.6%) and lower respiratory specimens (43%) (one-half were isolated from cystic fibrosis patients). Accurate genus vs species identities were obtained for 67.7% and 32.5% aerobic GNBs, 73% and 60% fGNCBs, and 23.5% CAMPB (with no discrepant species), respectively. Wrong or no results were obtained for 81 (32.3%) aerobic GNBs, 17 (27%) fGNCBs, and 13 (72.2%) CAMPB. No results or misidentifications occurred for 33% of aerobic GNBs, 26% of fGNCBs, and 76.5% of CAMPB due to absence of species in the instrument's database. VITEK MS performance remained stable for aerobic GNBs and fGNCBs but improved for CAMPB with addition of *Campylobacter rectus* and *Campylobacter curvus* to the database. 16S remains important for identification of GNOs when proteomics fails.

**IMPORTANCE** Matrix-assisted laser desorption ionization-time of flight mass spectrometry (MALDI-TOF MS) has transformed the identification of commonly encountered Gram-negative organisms (GNOs) in the clinical laboratory, but rare and unusual organisms continue to challenge the technology. This study verified performance of VITEK MS for identification of a broad range of rare and unusual clinical GNO isolates by our large reference laboratory workflow over a multi-year period. Although most GNOs were accurately identified by MALDI-TOF MS, a small number of clinical isolates (~1%–6%) required 16S sequencing for identification depending on the GNO category. Approximately one-third of aerobic Gram-negative bacilli (GNBs) and two-thirds of *Campylobacterales* could not be accurately identified by proteomics due to lack of an organism in the instrument's database. MALDI-TOF MS databases should be continuously updated and validated, and laboratories should have a workflow for identification of unusual or rarely encountered aerobic, fastidious, and *Campylobacterales* GNOs that

Address correspondence to D. L. Church, dchurch@ucalgary.ca.

The authors declare no conflict of interest.

includes 16S rRNA gene sequencing whenever proteomics cannot give a definitive identification.

**KEYWORDS** identification, Gram-negative bacilli, VITEK MS, proteomics

Gram-negative organisms (GNs) are commonly encountered pathogens in clinical microbiology laboratories. Non-fermenting Gram-negative bacteria (nfGNBs) are largely environmental opportunists causing severe infections in immunocompromised individuals (1–5). Pathogenic colonization in cystic fibrosis patients, especially with *Pseudomonas aeruginosa*, results in decline in pulmonary function and potential morbidity and mortality (6–8). Human gut microbiomes also include enteric Gram-negative bacilli (eGNBs), which cause urinary tract or intra-abdominal infections with associated bloodstream infection (1, 9, 10). Fastidious Gram-negative coccobacilli (fGNCBs) within the HACEK group (*Haemophilus*, *Aggregatibacter*, *Cardiobacterium*, *Eikenella*, and *Kingella*) are rare causes of infective endocarditis, but other fGNCBs cause various other types of infection (11, 12). *Campylobacterales* (CAMPB) cause gastrointestinal infections including peptic ulcer disease (i.e., *Helicobacter pylori*) but rarely cause other diseases (13–16). Effective therapy requires accurate identification of pathogenic GNOs due to their unique antibiotic susceptibility profiles, and inherent or developing multi-drug resistance (9, 17–19).

Definitive identification of GNOs is sub-optimal using phenotypic methods because of low biochemical reactivity. Accuracy and timeliness of GNO identification along with patient outcomes has improved with use of matrix-assisted laser desorption ionization-time of flight mass spectrometry (MALDI-TOF MS) (20, 21). MALDI-TOF MS databases currently lack spectral profiles for all clinically relevant unusual and rarely encountered pathogenic GNOs resulting in no isolate identification or misidentification. Limited studies compared VITEK MS (MALDI-TOF MS) (bioMérieux, Laval, Que.) to 16S for identification of unusual or rarely encountered pathogenic GNOs (22–25). Prior reports of MALDI-TOF MS performance used GNO isolates that had prior 16S analysis to include organisms available in proteomics databases (22–25). We verified performance of our laboratory's' workflow by comparing VITEK MS results since implementation in 2014 to secondary 16S analysis of isolates that proteomics gave no result or misidentified. We encounter a high number of unusual or rarely invasive GNO compared to smaller hospital laboratories in our regional network due to the consolidated regional nature of our large referral facility.

## MATERIALS AND METHODS

### Patients and clinical specimens

Patients who had ≥1 GNO isolate from blood or non-blood invasive clinical specimens were enrolled over a multi-year period (2014–2019) from Calgary and South Zones, Alberta Health Services (AHS) following clinical review by a medical microbiologist and Infectious Diseases specialist. Adults had two sets of blood cultures (i.e., each consisting of an aerobic/anaerobic bottle) according to Calgary Zone regional protocol. Non-blood specimens (abscesses, tissues, sterile fluids) were collected operatively or by interventional radiology under ultrasound guidance. Non-blood specimens were collected using standard collection devices and protocols to ensure recovery of GNB. Clinical specimens were transported for immediate processing to laboratory within 2 h after collection.

### Laboratory setting

Our large regional laboratory in Calgary Zone, Alberta Health Service does clinical microbiology testing for Calgary and surrounding hamlets and towns in Southern Alberta representing an urban and rural population of ~2.6 million people including adult tertiary hospitals, a tertiary pediatric hospital, southern rural facilities, all ambulatory practices, and long-term care centers.

## Laboratory analyses

Clinical specimens were initially analyzed using standard phenotypic methods including a microscopic examination and aerobic and anaerobic culture methods (except for sputa). Blood and sterile fluids were cultured using an aerobic BACT/ALERT FA plus and FN plus bottle pair incubated for up to 5 days in a BacT/Alert instrument (bioMérieux, Laval, Quebec). Subculture of positive blood cultures inoculated blood agar (BA), chocolate agar (CHOC), MacConkey agar (MAC), and Brucella blood agar (BBA) (Dalynn, Calgary) incubated for 5 days at 35°C. Subculture of sputa and or deep lung samples [i.e., bronchoalveolar lavages (BALs), bronchial washes (BWs)] inoculated BA, CHOC, MAC, and buffered charcoal yeast extract agar (BCYE for recovery of *Legionella*) incubated under aerobic conditions at 35°C for 4 days. Tissue and deep abscess specimens were inoculated to BA, CHOC, and BBA; BA and CHOC plates were incubated in 5% $CO_2$, MAC was incubated in $O_2$, and BBA was incubated in an Anoxomat anaerobic jar system (Fisher Scientific, Mississauga, Ontario) for 4 days. GNB isolates were confirmed as enteric, non-fermenting, fastidious, or microaerophilic by standard phenotypic procedures (i.e., growth on specific media, atmospheric growth conditions, colony morphology, Gram stain reaction/morphology, and rapid biochemical tests). Categories of GNOs included (i) enteric isolates were glucose and lactose fermenting *Enterobacterales* species (eGNBs), (ii) non-fermenting isolates were glucose and lactose non-fermenting species (nfGNBs), and (iii) fastidious Gram-negative coccobacilli isolates did not grow on MAC and/or BA (fGNCBs) and CAMPB. Although >99% of *Enterobacterales* isolates were accurately identified by VITEK MS, a few eGNBs ($n = 8$) that could only be accurately identified by 16S sequencing were included (Table 1).

All GNO isolates were subsequently analyzed according to our laboratory's workflow; MALDI-TOF MS is first done with subsequent 16S rRNA gene sequencing when proteomics gives no result of a low confidence. Direct deposit-full formic acid extraction was done for MALDI-TOF MS (VITEK MS, bioMérieux, Laval, Quebec) analysis according to manufacturer's instructions as described in previously published guidelines (27). Repeat MALDI-TOF MS was done with the same method if initial results gave low confidence (<99%) or no identification (i.e., no results). Reported VITEK MS results had a high confidence (i.e., ≥99.0%) and agreed with phenotypic results. VITEK MS V.2.0 (2012) database was used from 2014 to 2016, V.3.0 from 2016 to 2018, and V.3.2 until 2019. A total of 14,478 (GNOs) including ~8.945 eGNBs, ~4,500 nfGNBs, ~978 fGNBs, and 55 *Campylobacterales* isolates were not enrolled because they were routinely identified with high confidence using phenotypic plus VITEK MS analysis and thus 16S was not performed. These were designated as "common GNOs" for the study. Enrolled GNOs all had molecular analysis because there was discrepancy between the phenotypic and proteomics results or proteomics analysis gave no result or a wrong identification. These were designated as "rare or unusual GNOs" for the study.

16S rRNA gene sequencing was performed by fast PCR/cycle sequencing using fast MicroSEQ 500 16S DNA PCR kits and an ABI Prism 3500 XL sequencer (Applied Biosystems, ThermoFisher Scientific, Foster City, CA) as previously described (28). SmartGene's Integrated Database Network System (IDNS3) (Lausanne, Switzerland) bacterial database was used to determine closely related species (https://www.Smartgene.com). Isolate sequences were compared to a well-characterized reference sequence and overall identity scores were 99.9% (0–2 mismatches). 16S rRNA gene sequencing identification to genus or species level used interpretive criteria outlined in Clinical and Laboratory Standards Institute (CLSI), Approved Guidelines MM-18 for targeted DNA sequencing analysis (26).

## Data analysis

Data were entered into a Microsoft Excel spreadsheet (MS Office 2016) and analyzed according to standard descriptive methods. VITEK MS performance was calculated against the current instrument databases' (i.e., V.2.0, V.3.0, or V.3.2) ability to accurately

**TABLE 1** Performance of VITEK MS compared to molecular identification of aerobic Gram-negative bacilli[a]

| Reference method[b] 16S RNA gene sequencing result[f] | VITEK MS results[b] | | | | | Total |
| --- | --- | --- | --- | --- | --- | --- |
| | No. (%) correct to genus | No. (%) correct to genus and species[e] | No. (%) with discordant species result[c] | No. (%) with wrong result[c] | No. (%) with no result | Total no. (%) |
| Achromobacter xyloxidans | 27 | | 27 (Achromobacter xyloxidans/denitrificans)[c] | | | 27 |
| Achromobacter xyloxidans/denitrificans split | 1 | | 1 (Achromobacter xyloxidans/denitrificans)[c] | | | 1 |
| Achromobacter sp. | 2 | – | | | | 2 |
| Acinetobacter junii | 2 | 2 | | | | 2 |
| Acinetobacter nosocomialis | 1 | 1 | | | | 1 |
| Acinetobacter sp. | 1 | – | 1 (Acinetobacter junii/haemolyticus) | | 1 | 2 |
| Alcaligenes faecalis | 2 | 2 | | | | 2 |
| Alcaligenes sp. | 2 | – | 1 (Alcaligenes faecalis) | 1 (Achromobacter xyloxidans/denitrificans split) | 2 | 5 |
| **Aureimonas altamirensis** | | | | | 1 | 1 |
| **Azorhizobium caulinodans** | | | | | 1 | 1 |
| **Bacteroidetes bacterium Sm46** | | – | | | 1 | 1 |
| Bergeyella zoohelcum | 2 | 2 | | | | 2 |
| Bergeyella sp. | 1 | – | 1 (Bergeyella zoohelcum) | | | 1 |
| Bordetella bronchiseptica | 1 | 1 | | | | 1 |
| Bordetella hinzii | 1 | 1 | | | | 1 |
| Bordetella holmesii | 1 | – | 1 (Bordetella trematum) | | 1 | 2 |
| **Bordetella petrii** | 1 | – | 1 (Bordetella avium) | 3 (Achromobacter xyloxidans/denitrificans split) | 1 | 5 |
| Brevibacillus agri | 1 | | | | | 1 |
| Brevibacillus brevis | 2 | | | | | 2 |
| Burkholderia cepacia complex | 23 | 1 | 22 [Burkholderia multivorans(16); B. cenocepacia (3); B. cepacia/B. vietnamiense split(3)][b] | | 1 | 24 |
| Burkholderia contaminans | 4 | | 4 [Burkholderia multivorans (3); B. cepacia (1)] | | | 4 |
| Burkholderia gladioli | 4 | 4 | | | | 4 |
| Burkholderia multivorans | 10 | 10 | | | 1 | 11 |
| Burkholderia pseudomultivorans | 6 | | 6 [Burkholderia cenocepacia(2); B. multivorans(3); B. ambifaria (1)] | | | 6 |
| Burkholderia seminalis | 2 | | 2 (Burkholderia multivorans) | | | 2 |
| Burkholderia sp. | 4 | – | | | | 4 |
| **Cellulomonas sp. MBEB32** | | – | | 1 (Microbacterium flavescens) | | 1 |
| **Chryseobacterium aquifrigidense** | 1 | | 1 (Chryseobacterium gleum) | | | 1 |
| **Chryseobacterium hominis** | | | | | 1 | 1 |
| Chryseobacterium indologenes | 1 | 1 | | | | 1 |
| **Chryseobacterium profundimaris** | | | | | 1 | 1 |
| Chryseobacterium sp. | 3 | – | 1 (Chryseobacterium indologenes) | 1 (Pleisiomonas shigelloides) | 2 | 6 |

TABLE 1 Performance of VITEK MS compared to molecular identification of aerobic Gram-negative bacilli[a] (Continued)

| Reference method[b] | VITEK MS results[b] | | | | | Total |
| 16S RNA gene sequencing result[f] | No. (%) correct to genus | No. (%) correct to genus and species[e] | No. (%) with discordant species result[c] | No. (%) with wrong result[c] | No. (%) with no result | Total no. (%) |
|---|---|---|---|---|---|---|
| **Chryseobacterium sp. DY46** | | — | | | 2 | 2 |
| Comamonas kerstersii | 1 | | 1 (Comomonas aquatica) | | | 1 |
| Comamonas testosteroni | 1 | 1 | | | | 1 |
| Cupriavidus gilardii | 1 | 1 | | | | 1 |
| Cupriavidus pauculus | | | | | 2 | 2 |
| Cupriavidus respiraculi | 2 | 2 | | | 2 | 4 |
| Cupriavidus sp. | 1 | — | 1 (Cupriavidus pauculus) | | | 1 |
| Delftia acidovorans 485 | 1 | 1 | | | | 1 |
| Delftia tsuruhatensis | 1 | | 1 (Delftia acidovorans) | | | 1 |
| **Dickeya chrysanthemi** | | | | | 1 | 1 |
| Elizabethkingia miricola | 2 | 2 | | | | 2 |
| **Enhydrobacter aerosaccus** | | | | 3 (Moraxella osloensis/E. aerococcus) | | 3 |
| Enterobacter sp. | | — | | 2 (Leclercia adecarboxylata) | | 2 |
| Escherichia coli | 1 | 1 | | 2 (Acinetobacter haemolyticus) | 1 | 4 |
| Hafnia paralvei | 1 | 1 | 1 (Hafnia alvei/Obesumbacterium proteus) | | | 1 |
| Herbaspirillum huttiense | | | | 1 (Burkholderia cepacia complex) | | 1 |
| Herbaspirillum sp. | | — | | 1 (Burkholderia cepacia complex) | | 1 |
| **Ignatzschineria indica** | | | | | 1 | 1 |
| Inquilinus limosus | | | | 2 (Sphingomonas paucimobilis or Micrococcus luteus/lylae) | 3 | 5 |
| **Kaistella flava** | | | | | 1 | 1 |
| **Kersteria gyiorum** | | | | | 1 | 1 |
| Kluyvera ascorbata | 1 | 1 | | | | 1 |
| Legionella bozemanii | 1 | 1 | | | 1 | 2 |
| **Legionella maceachernii** | | | | | 1 | 1 |
| Legionella pneumophila | 7 | 7 | | | | 7 |
| **Massilia brevitalea** | | | | | 1 | 1 |
| **Massilia oculi** | | | | | 1 | 1 |
| Ochrobactrum anthropi | 1 | 1 | | | | 1 |
| Ochrobactrum sp. | 1 | — | 1 (O. intermedium) | | 1 | 2 |
| Odoribacter splanchnicus | | | | | 1 | 1 |
| Odoribacter sp. | | — | | | 1 | 1 |

**TABLE 1** Performance of VITEK MS compared to molecular identification of aerobic Gram-negative bacilli[a] (Continued)

| Reference method[b] 16S RNA gene sequencing result[f] | VITEK MS results[b] | | | | | Total |
| --- | --- | --- | --- | --- | --- | --- |
| | No. (%) correct to genus | No. (%) correct to genus and species[e] | No. (%) with discordant species result[c] | No. (%) with wrong result[c] | No. (%) with no result | Total no. (%) |
| *Oligella uli* | | | | | 1 | 1 |
| *Oligella urethralis* | 4 | 4 | | | | 4 |
| **Pantoea gaviniae** | | | | | 1 | 1 |
| *Pantoea* sp. | | – | | | 1 | 1 |
| **Paracoccus panacisoli** | | | | 1 (*Oligella ureolytica*) | 1 | 2 |
| **Paracoccus sp. KS-11 Z8B-43** | | – | | 1 (*Methylobacterium fujisawaense*) | | 1 |
| *Paracoccus yeei* | 7 | 7 | | | | 7 |
| *Paracoccus* sp. | | – | | | 2 | 2 |
| **Parapusillimonas granuli** | | | | | 1 | 1 |
| **Pedobacter suwonensis** | | | | | 1 | 1 |
| *Pseudomonas aeruginosa* | | | | | 1 | 1 |
| *Pseudomonas alcaligenes* | 1 | 1 | | | | 1 |
| **Pseudomonas costantinii** | | | | 1 (*Pseudomonas veronii*) | | 1 |
| **Pseudomonas moraviensis** | | | | 1 (*Pseudomonas fluorescens*) | | 1 |
| **Pseudomonas nitroreducens** | | | | | 1 | 1 |
| **Pseudomonas trivialis** | 1 | | 1 (*Pseudomonas veronii*) | | | 1 |
| *Pseudomonas* sp. | 2 | – | 2 (*Pseudomonas orzyihabitans*) | | | 2 |
| *Psychrobacter phenylpyruvicus* | 3 | 3 | | | | 3 |
| **Psychrobacter sanguinis** | 2 | | 2 (*Psychrobacter phenylpyruvicus*) | | | 2 |
| *Psychrobacter* sp. | 1 | – | 1 (*P. phenylpyruvicus*) | | | 1 |
| *Rahnella aquatilis* | 1 | 1 | | | | 1 |
| *Rahnella* sp. | 1 | – | 1 (*R. aquatilis*) | 1 (*Ewingella americana*) | | 2 |
| *Ralstonia pickettii* | 1 | | 1 (*Ralstonia mannitolilytica*) | | | 1 |
| *Ralstonia* sp. | 1 | – | 1 (*Ralstonia pickettii*) | | | 1 |
| *Rhizobium* sp. | 1 | – | 1 (*Rhizobium radiobacter*) | 1 (*Ochrobactrum anthropi*) | | 2 |
| **Roseomonas gilardii** | 1 | 1 | 1 (*Roseomonas mucosa*) | | | 2 |
| *Roseomonas mucosa* | 1 | 1 | | 1 (*Pseudomonas fluorescens*) | | 2 |
| *Roseomonas* sp. | 2 | – | 2 (*Roseomonas gilardii*) | 1 (*Bordetella bronchiseptica*) | 1 | 4 |
| **Sneathia amnii** | | | | 1 (*Brevundimonas diminuta*) | 2 | 3 |
| *Sphingomonas ginsenosidimutans* | 1 | | 1 (*Sphingomonas adhaesiva*) | | | 1 |
| *Sphingomonas paucimobilis* | 2 | 2 | | | | 2 |
| *Sphingomonas* sp. | 3 | – | 1 (*Sphingomonas melonsis*) | | | 3 |
| **Sphingobacterium spiritivorum** | | | | | 2 | 2 |
| *Stenotrophomonas maltophilia* | | | | | 2 | 2 |
| *Stenotrophomonas* sp. | | – | | | 1 | 1 |

**TABLE 1** Performance of VITEK MS compared to molecular identification of aerobic Gram–negative bacilli[a] (Continued)

| Reference method[b] 16S RNA gene sequencing result[f] | VITEK MS results[b] | | | | | Total Total no. (%) |
|---|---|---|---|---|---|---|
| | No. (%) correct to genus | No. (%) correct to genus and species[e] | No. (%) with discordant species result[c] | No. (%) with wrong result[c] | No. (%) with no result | |
| *Tissierella carlieri* | | | | | 1 | 1 |
| *Vibrio cholerae* | 1 | 1 | | | | 1 |
| *Vibrio fluvialis* | 1 | 1 | | | | 1 |
| *Vibrio harveyi* | 1 | 1 | | | | 1 |
| *Vibrio parahaemolyticus* | 1 | | 1 (*Vibrio alginolyticus*) | | | 1 |
| *Yersinia aleksiciae* | | | | | 1 | 1 |
| *Yersinia pestis/Yersina pseudotuberculosis* | 1 | | 1 (*Yersinia pseudotuberculosis*) | | | 1 |
| Total | 170/251 (67.7%)[d] | 65/170 (38.2%)[d] 65/200 (32.5%)[e] | 91/170 (53.5%)[d] | 26/251 (10.4%) | 55/251 (22%) | 251 |

[a]A few eGNBs (*n* = 8) were enrolled that could not be identified by phenotypic methods or proteomics (pale blue boxes).
[b]All isolates were analyzed by both VITEK MS and 16S rRNA gene sequencing as the reference method. Proteomics and genomic results were compared to phenotypic results (i.e., Gram stain and phenotypic identification).
[c]Species not included or limited species representation in the VITEK MS databases: V.2.0 (used 2012–2014), V.3.0 (used 2016–2017), and V.3.2(2018–2019) are highlighted in bold; none of these specific genera and/or species were in the database in use at the time of proteomics analysis and had not been added as of V.3.2. VITEK MS does not distinguish between *Achromobacter xyloxidans/denitrificans* or separate species within the *Burkholderia cepacia* complex.
[d]Accurate genus and species identification based on the inclusion of the organism in the currently used version of the VITEK MS database.
[e]Accurate genus and species identifications based on the total number of clinically relevant aerobic non-fermenting and enteric GNBs studied. The total isolates scored for species-level identification by proteomics were decreased in this column if 16S only provided a genus-level identification and these are indicated by dashes.
[f]Based on a multi-sequence alignment using a reference strain and review of the molecular species identification according to the CLSI MM-18 Guidelines (26).

identify GNBs at the time of analysis. VITEK MS performance was compared to the "gold standard" method (i.e., 16S rRNA gene sequencing). Different categories of GNB results were analyzed separately according to Tables 1 to 3. The total isolates scored for accurate species-level identification by proteomics were decreased by the number of isolates for which 16S only provided a genus-level identification.

## RESULTS

### Clinical specimens and isolates

Patients' diagnosis included bloodstream infection, pneumonia, lung abscesses, cystic fibrosis, deep wound infections, deep organ and intra-abdominal abscesses, and deep skin and soft tissue infections including burns. Three hundred thirty-one GNO isolates were recovered from 304 specimens; >1 isolate was recovered from 6% ($n = 19$) specimens. Isolates studied included 243 (73.4%) nfGNBs and 8 (2.4%) eGNBs (Table 1), 63 (19%) fGNCBs (Table 2), and 17 (5.1%) CAMPB (Table 3). This represents ~5.4% of nfGNBs, <1% of eGNBs, ~6.3% of fGNCBs, and ~31% of CAMPB of the total number of each category of GNOs analyzed by the clinical laboratory workflow during the study period. The recovery of isolates from clinical specimens is outlined in Fig. 1; 75% were recovered from either blood cultures (31.6%) or lower respiratory site/sources (43%) including BALs, BWs, and sputum and throat specimens from cystic fibrosis. One-half of clinical respiratory specimens were submitted from patients with cystic fibrosis. Enrollment of isolates remained constant with an average of 59 (range = 41 to 89) isolates accrued each per year (data not shown).

### Description of GNOs

Table 1 describes 37 genera and 80 species of aerobic GNBs studied. Predominant genera were *Achromobacter* spp. 12% ($n = 28$) and *Burkholderia* spp. 22% ($n = 53$), but a diverse spectrum of aerobic GNBs were represented. Predominant species included *Achromobacter xyloxidans* ($n = 25$), *Bordetella* spp. ($n = 9$) (*Bordetella bronchiseptica*, *Bordetella hinzii*, *Bordetella holmesii*, and *Bordetella petrii*), *Chryseobacterium* spp. ($n = 12$) (*Chryseobacterium aquifrigidense*, *Chryseobacterium hominis*, *Chryseobacterium indologenes*, and *Chryseobacterium profundimaris*), *Legionella* spp. ($n = 10$) (*Legionella bozemanae*, *Legionella maceachernii*, and *Legionella pneumophila*), *Paracoccus* spp. ($n = 12$) (*Paracoccus panacisoli* and *Paracoccus yeei*), *Pseudomonas* spp. ($n = 8$) (*P. aeruginosa*, *Pseudomonas alcaligenes*, *Pseudomonas costantinii*, *Pseudomonas moraviensis*, *Pseudomonas nitroreducens*, and *Pseudomonas trivialis*), and *Sphingomonas* spp. (*Sphingomonas paucimobilis* and *Sphingomonas spiritivorum*) (Table 1). Accuracy of VITEK MS aerobic GNB identification remained stable due to limited addition of new species to the instrument's spectral database profiles; all organisms missing from the instrument's database are highlighted in Table 1 in bold.

The 9 genera and 25 species of fGNCBs studied are outlined in Table 2. Predominant genera were *Aggregatibacter* spp. 21% ($n = 13$), *Capnocytophaga* spp. 16% ($n = 10$), and *Neisseria* spp. 18% ($n = 11$) but a diverse spectrum of fGNCBs were represented. Predominant species included *Aggregatibacter* spp. ($n = 16$) (*Aggregatibacter actinomycetemcomitans*, *Aggregatibacter aphrophilus*, *Aggregatibacter kilianii*, and *Aggregatibacter segnis*), *Capnocytophaga* spp. ($n = 13$) (*Capnocytophaga canis*, *Capnocytophaga cynodegmi*, and *Capnocytophaga sputigena*), and *Neisseria* spp. ($n = 11$) (*Neisseria elongata*, *Neisseria flavescens*, *Neisseria meningitidis*, *Neisseria mucosa*, *Neisseria subflava*, and *Neisseria weaveri*). Accuracy pf VITEK MS fGNCB identification remained stable due to limited addition of new fGNCB spectral database profiles; all organisms missing from the instrument's database are highlighted in Table 2 in bold.

The 2 genera [*Campylobacter* spp. 65% ($n = 11$) and *Helicobacter* spp. 35% ($n = 6$)] and 11 species of CAMPB studied are outlined in Table 3. Predominant species included *Campylobacter gracilis*, *Campylobacter ureolyticus*, and *Helicobacter cinaedi*. Accuracy of VITEK MS CAMPB identification improved using V.3.2 database with the

**TABLE 2** Performance of VITEK MS for identification of fastidious Gram-negative coccobacilli

| Reference method[a] 16S RNA gene sequencing results[e] | VITEK MS results[a] | | | | | Total |
|---|---|---|---|---|---|---|
| | No. (%) correct to genus | No. (%) correct to genus and species[d] | No. (%) with discordant species results[b] | No. (%) with wrong result[b] | No. (%) with no results | Total no. (%) |
| *Aggregatibacter actinomycetemcomitans* | 4 | 4 | | | | 4 |
| *Aggregatibacter aphrophilus* | 6 | 5 | 1 (*Aggregatibacter. segnis*) | 1 (*Nocardia* sp.) | | 7 |
| ***Aggregatibacter kilianii*** | | | | 1 (*H. parainfluenzae*) | | 1 |
| *Aggregatibacter segnis* | 2 | 2 | | | | 2 |
| *Aggregatibacter* sp. | 2 | – | 2 [*A. segnis*(1); *A. segnis/aphrophilus* split(1)] | | | 2 |
| *Capnocytophaga canimorsus* | 5 | 5 | | | 2 | 7 |
| ***Capnocytophaga canis*** | | | | | 1 | 1 |
| ***Capnocytophaga cynodegmi*** | 1 | | 1 (*C. canimorsus*) | | | 1 |
| *Capnocytophaga sputigena* | 4 | 4 | | | | 4 |
| *Cardiobacterium hominis* | 3 | 3 | | | | 3 |
| *Cardiobacterium* sp. | | – | | | 1 | 1 |
| *Haemophilus haemoglobinophilus* | 1 | 1 | | | | 1 |
| *Haemophilus parainfluenzae* | | | | | 1 | 1 |
| *Haemophilus* sp. | 1 | – | 1 (*H. haemolyticus*) | | | 1 |
| *Kingella kingae* | 1 | 1 | | | | 1 |
| ***Kingella negevensis*** | 1 | | 1 (*K. kingae*) | | | 1 |
| ***Kingella oralis*** | | | | 1 (*Vibrio parahaemolyticus*) | | 1 |
| *Moraxella catarrhalis* | 1 | 1 | | | | 1 |
| ***Moraxella osloensis*** | | | | 3 (*M. osloensis/Enhydrobacter aerococcus*) | | 3 |
| *Moraxella* sp. | | – | | | 1 | 1 |
| *Neisseria elongata* | 3 | 3 | | | | 3 |
| ***Neisseria flavescens*** | 1 | | 1 (*N. subflava*) | | | 1 |
| *Neisseria meningitidis* | 2 | 2 | | | | 2 |
| *Neisseria mucosa* | 1 | 1 | | | | 1 |
| *Neisseria subflava* | 1 | | 1 (*Neisseria* sp. split) | | | 1 |
| *Neisseria weaveri* | 2 | 1 | 1 (*N. zoodegmatis*) | | 1 | 3 |
| *Neisseria* sp. | 1 | – | 1 (*N. polysaccharea*) | | | 1 |
| *Pasteurella canis* | 3 | | 3 (*Pasteurella stomatis*) | | | 3 |
| *Pasteurella dagmatis* | | | | | 1 | 1 |
| ***Pasteurella* sp. MCCM 02120** | | – | | 1 (*Mannhaemia haemolytica*) | 1 | 2 |
| ***Pasteurellaceae bacterium*** | | | | | 1 | 1 |
| Total | 46/63 (73%)[d] | 33/46 (71.7%)[c] 33/55 (60%)[d] | 13/46 (28.3%)[e] | 7/63 (11.1%) | 10/63 (15.9%) | 63 |

[a]All isolates were analyzed by both VITEK MS and 16S rRNA gene sequencing as the reference method. Proteomics and genomic results were compared to phenotypic results (i.e., Gram stain and phenotypic identification).

[b]Species not included or limited species representation in the VITEK MS databases: V.2.0 (used 2012–2014), V.3.0 (used 2016–2017) and V.3.2(2018–2019) are highlighted in bold; none of these specific genera and/or species were in the database in use at the time of proteomics analysis and had not been added as of V.3.2. VITEK MS does not distinguish between *Moraxella osloensis/Enhydrobacter*.

[c]Accurate genus- and species-level identification based on the inclusion of the organism in the currently used version of the VITEK MS database.

[d]Accurate genus- and species-level identifications based on the total number of clinically relevant aerobic non-fermenting and enteric GNBs studied. The total isolates scored for species-level identification by proteomics is decreased in this column if 16S only provided a genus-level identification and these are indicated by dashes.

[e]Based on a multi-sequence alignment using a reference strain and review of the molecular species identification according to the CLSI MM-18 Guidelines (26).

addition of *Campylobacter curvus* and *Campylobacter rectus*; all organisms missing from the instrument's database are highlighted in Table 3 in bold.

**TABLE 3** Performance of VITEK MS for identification of Campylobacterales

| Reference method[a] 16S RNA gene sequencing results[e] | VITEK MS results[a] | | | | | Total |
|---|---|---|---|---|---|---|
| | No. (%) correct to genus | No. (%) correct to genus and species | No. (%) with discordant species results[b] | No. (%) with wrong results[b] | No. (%) with no results[b] | Total no. (%) |
| *Campylobacter concisus* | | | | | 1 | 1 |
| *Campylobacter curvus* | | | | 1 (*Enterococcus faecalis*) | | 1 |
| *Campylobacter gracilis* | | | | | 2 | 2 |
| *Campylobacter hyointestinalis* | 1 | 1 | | | | 1 |
| *Campylobacter rectus* | | | | 1 (*Paracoccus yeei*) | | 1 |
| *Campylobacter rectus/curvus* | | | | | 1 | 1 |
| *Campylobacter ureolyticus* | 2 | 2 | | | 1 | 3 |
| *Campylobacter* sp. oral taxon | | –[d] | | 1 (*Porphyromonas gingivalis*) | | 1 |
| *Helicobacter bilis* | | | | | 1 | 1 |
| *Helicobacter cinaedi* | | | | | 2 | 2 |
| *Helicobacter pullorum* | | | | 1 (*Klebsiella pneumoniae*) | | 1 |
| *Helicobacter pylori* | 1 | 1 | | | | 1 |
| *Helicobacter rappini* | | | | | 1 | 1 |
| Total | 4/17 (23.5%)[d] | 4/4 (100%)[c] 4/16 (25%)[d] | None | 4/17 (23.5%) | 9/17 (52.9%) | 17 |

[a]All isolates were analyzed by both VITEK MS and 16S rRNA gene sequencing as the reference method. Proteomics and genomic results were compared to phenotypic results (i.e., Gram stain and phenotypic identification).

[b]Species not included or limited species representation in the VITEK MS databases: V.2.0 (used 2012–2014), V.3.0 (used 2016–2017), and V.3.2 (2018–2019) are highlighted in bold; none of these specific genera and/or species were in the database in use at the time of proteomics analysis and had not been added as of V.3.2.

[c]Accurate genus- and species-level identification based on the inclusion of the organism in the currently used version of the VITEK MS database.

[d]Accurate genus- and species-level identifications based on the total number of clinically relevant aerobic non-fermenting and enteric GNBs studied. The total isolates scored for species-level identification by proteomics is decreased in this column if 16S only provided a genus-level identification and these are indicated by dashes.

[e]Based on a multi-sequence alignment using a reference strain and review of the molecular species identification according to the CLSI MM-18 Guidelines (26).

## Performance of VITEK MS for GNO identification

16S accurately identified 331 (100%) and 271 (81.9%) to the genus and species, respectively.

Overall, VITEK MS accurately identified 66.5% ($n$ = 220) and 30.8% ($n$ = 102) of GNOs to the genus and species, respectively. Wrong identifications and no results occurred for 37 (11.2%) and 74 (22.4%) isolates, respectively.

16S accurately identified 100% of isolates to genus and most 79.7% ($n$ = 200) of aerobic GNB species. Performance of VITEK MS compared to 16S for aerobic GNB identification is shown in Table 1. Proteomics accurately identified 67.7% and 32.5% isolates to genus and species, respectively; species-level identification scoring excluded isolates with only a genus-level identification by 16S. Discordant species-level identifications are also shown in Table 1. No result (22%) ($n$ = 55) or a wrong identity (10.4%) ($n$ = 26) occurred for 33% of aerobic GNBs because organisms were not included in the VITEK MS databases at the time of analysis (highlighted in bold in Table 1). Wrong identifications included *Microbacterium flavescens*, *Pleisiomonas shigelloides*, *Acinetobacter haemolyticus*, *Burkholderia cepacia* complex, *Sphingomonas paucimobilis*, *Micrococcus luteus/lylae* 50/50 split, *Oligella ureolytica*, *Methylobacterium fujisawaense*, *Ewingella americana*, *Ochrobactrum anthropi*, *Bordetella bronchiseptica*, and *Brevundimonas diminuta* (Table 1). VITEK MS does not distinguish between *Achromobacter xyloxidans/denitrificans* 50/50 split, *Moraxella osloensis/Enhydrobacter aerococcus* 50/50 split, and *Yersinia pseudotuberculosis/pestis*.

16S accurately identified 100% of isolate to genus and most 61.9% ($n$ = 39) fGNCB species. Performance of VITEK MS to 16S for fGNCB identification is shown in Table 2. Proteomics accurately identified 73% and 60% isolates to genus and species, respectively; species-level identification scoring excluded isolates with only a genus-level identification by 16S. Discordant species-level identifications are also shown in Table 2. No results (15.9%) ($n$ = 10) or a wrong identity (11.1%) ($n$ = 7) occurred for 26% of fGNCBs because organisms were not included in the VITEK MS database used for analysis

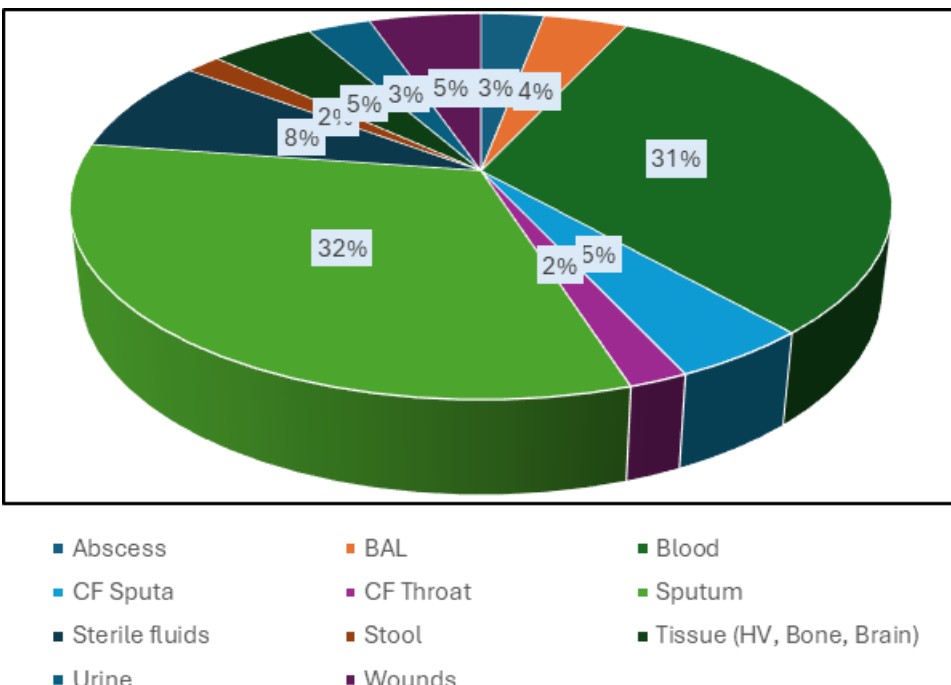

FIG 1   Recovery of GNOs from clinical specimens.

(highlighted in bold in Table 2). Wrong identifications included *Nocardia* sp., *Haemophilus parainfluenzae*, *Vibrio parahaemolyticus*, *Moraxella osloensis*, and *Mannhaemia haemolytica*.

16S accurately identified 100% of isolates to genus and most 94.1% (*n* = 16) CAMPB species. Performance of VITEK MS to 16S for CAMPB identification is shown in Table 3. Proteomics accurately identified 23.5% (*n* = 4) of *Campylobacterales* isolates; species-level identification was slightly better (25%) when excluding isolates with only a genus-level identification by 16S. No discordant species-level identifications occurred. No results or wrong identity occurred for most 76.5% (*n* = 13) CAMPB. No results (52.9%) (*n* = 9) or a wrong identity (23.5%) (*n* = 4) occurred for most CAMP because organisms were not included in VITEK MS database at the time of analysis (highlighted in bold in Table 3). Wrong identifications included *Enterococcus faecalis*, *Paracoccus yeei*, *Porphyromonas gingivalis*, and *Klebsiella pneumoniae*.

## DISCUSSION

This study uniquely highlights the utility and pitfalls of proteomics for accurate identification of rare and unusual GNBs within a sequential clinical microbiology workflow. Although most clinically encountered GNOs were accurately identified during the study period (data not shown), MALDI-TOF MS performance for study organisms was less optimal. VITEK MS gave an accurate genus result for 66.3% of GNOs but gave a wrong identification or no results for 14% and 22.4% of isolates, respectively. VITEK MS has difficulty identifying most *Campylobacterales* due to the limited number of *Campylobacter* and *Helicobacter* species available in the instrument's database. Addition of *Campylobacter curvus* and *C. rectus* to the V.3.2 database improved identification of these species in the later part of the study. Performance remained stable for most other types of GNOs since few organisms were added to instrument's database during the study. 16S provided an accurate genus identification for all isolates, but species identification could improve with sequencing of a much larger portion of the gene up to ~1,160 base pairs; many GNOs have almost complete homology within the 16S V1–V3

gene regions (~first 500 bp interrogated by the fast MicroSEQ 500 16S DNA PCR kits used) (26).

Previously reported MALDI-TOF MS verification studies showed improved proteomics performance compared to our data because of primary selection of isolates based on inclusion in an instruments' database. Faron and colleagues performed a large multi-center evaluation of Bruker MALDI Biotyper for identification of 2,263 NGNB isolates representing 23 genera and 61 species (29). Compared to sequencing, proteomics correctly identified 99.8% (2,258/2,263) to genus and 98.2% (2,222/2,269) to species (29). Gautam and colleagues analyzed 150 nfGNBs (either *Acinetobacter baumannii*, *Burkholderia cepacia* complex, or *Stenotrophomonas maltophilia*) isolates using MALDI Biotyper (Bruker Daltronics) following molecular confirmation and found genus and species identification agreement of 100% and 73.3%, respectively (30).

Other studies focused on performance of MALDI-TOF MS for identification of respiratory tract nfGNBs isolated from cystic fibrosis patients (7, 25, 31, 32). Plonga and colleagues assessed VITEK MS for this purpose and showed a 58.5% and 46.2% accuracy for genus compared to species identification using V.2.0 to 3.2 databases (7). Accuracy improved to 89.2% and 83.6% for genus compared to species identification using the instrument's SARAMIS database (research use only) due to improved identification of *Burkholderia cenocepacia* and *Burkholderia contaminans*. Few reports compare proteomics for accurate identification of fGNBs. Branda and colleagues performed a multicenter validation of VITEK MS (V.2.0 database) including 226 isolates representing of 89 genera and 15 species of previously sequenced fGNBs; 96% of isolates were accurately identified to genus or species (22). Schulthess and colleagues studied 150 nfGNBs and 50 fGNBs previously characterized (i.e., phenotypic methods and 16S) isolates using MALDI Biotyper, and found lower accuracy for genus (53.7%) and species (46.4%) identification despite performing direct transfer plus formic acid preparation and ethanol-formic acid extraction (33). Proteomics performance for *Campylobacterales* has not previously been reported.

Approximately one-third of proteomic errors for nfGNBs and fGNCBs and two-thirds of CAMPB in our study occurred because specific organisms were not included in VITEK MS databases at the time of isolate analysis (Tables 1 to 3, highlighted in bold). But proteomics errors have also been reported to occur because a quality peak is difficult to distinguish with specific spectra (34). MALDI-TOF MS may also have difficulty separating specific GNB genus and species giving lower quality identification scores related to interspecies similarities (26, 27). Clinically important GNOs may also have closely related 16S genetic sequences and proteomics spectral profiles decreasing these methods' ability to give an accurate identification (35). 16S gene sequencing should remain in place within our laboratory's workflow until sufficient proteomic spectra for rare and unusual nfGNBs, fGNCBs, and *Campylobacterales* are included in the VITEK MS instrument databases.

Limitations of this work include enrollment of isolates from a single Canadian center albeit our laboratory performs all microbiology testing for an entire regional population. Geographic strain differences for GNOs may cause discordant identification rates, and limited overall data comparing proteomics results across MALDI-TOF MS platforms have been globally reported. Chart reviews may have improved assessment of pathogenicity of these isolates, but each case was medically assessed prior to enrollment. A caveat to clinical identification accuracy remains necessity of reporting a genus versus species; although only genus identity may suffice for patient care, rare and unusual GNO species have unique inherent or acquired multi-drug resistance profiles and epidemiologic patterns of disease that may necessitate an accurate species identification.

## Conclusions

Laboratories should have a workflow for identification of pathogenic unusual or rarely encountered aerobic, fastidious, and *Campylobacterales* GNOs that includes 16S rRNA gene sequencing whenever proteomics cannot give a definitive identification.

## ACKNOWLEDGMENTS

We would like to thank medical laboratory technologists at Alberta Precision Laboratories for their assistance with isolate testing.

D.L. Church: formal analysis, conceptualization, data curation, methodology, project administration, writing – original draft, writing – review and editing. T. Griener: methodology, writing – review and editing. D. Gregson: methodology, writing – review & editing.

All authors declare this research was conducted without any commercial or financial relationships that could be construed as a potential conflict of interest.

## AUTHOR AFFILIATIONS

[1]Department of Pathology & Laboratory Medicine, Cummings School of Medicine, University of Calgary, Calgary, Canada
[2]Department of Medicine, Cummings School of Medicine, University of Calgary, Calgary, Canada
[3]Alberta Precision Laboratories, Calgary, Canada

## AUTHOR ORCIDs

D. L. Church http://orcid.org/0000-0002-8306-8099

## AUTHOR CONTRIBUTIONS

D. L. Church, Conceptualization, Data curation, Formal analysis, Methodology, Writing – original draft, Writing – review and editing | T. Griener, Formal analysis, Methodology, Validation | D. Gregson, Formal analysis, Methodology, Writing – review and editing

## DATA AVAILABILITY

Data support findings of this study are available from Alberta Health Services (AHS), Alberta Precision Laboratories (APL) but restrictions apply to availability of se data, were used under ethics agreement for current study, and so are not publicly available. Data are available from authors upon reasonable request and with permission of AHS/APL.

## ETHICS APPROVAL

This study was approved, and a waiver of consent was granted by Conjoint Health Ethics Research Board (CHREB), Alberta Health Services, and University of Calgary (REB15-0629). This study complied with all relevant guidelines and regulations.

## ADDITIONAL FILES

The following material is available online.

### Open Peer Review

**PEER REVIEW HISTORY (review-history.pdf).** An accounting of the reviewer comments and feedback.

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
