## [Reviewer comments · Microbiology Spectrum]

Microbiology Spectrum

Multi-Year Comparison of VITEK® MS performance for identification of rarely encountered pathogenic gram-negative bacilli (GNBs) in a large integrated Canadian healthcare region.

Deirdre Church, Thomas Griener, and Dan Gregson

Corresponding Author(s): Deirdre Church, University of Calgary Cumming School of Medicine

Review Timeline:

Submission Date:	September 12, 2024
Editorial Decision:	September 19, 2024
Revision Received:	September 28, 2024
Accepted:	October 1, 2024

Editor: Karen Carroll

Reviewer(s): The reviewers have opted to remain anonymous.

Transaction Report:

DOI: <https://doi.org/10.1128/spectrum.02276-24>

Re: Spectrum02276-24 (Multi-Year Comparison of VITEK® MS performance for identification of rarely encountered pathogenic gram-negative bacilli (GNBs) in a large integrated Canadian healthcare region.)

Dear Dr. Deirdre L. Church:

Thank you for the privilege of reviewing your work. I have reviewed your responses to the reviewers' comments and most of the issues have been addressed. However, before we can accept the paper, we would like to have you go over it again and address the minor issues I have highlighted here:

- 1) Please define the abbreviations in the abstract. Alternatively, spelling them out here and defining them in the body of the text is preferred. That said, please make sure you are consistent with abbreviations throughout the text. For example, glucose non-fermenting Gram-negative bacilli are abbreviated as nFGNB, NFGNB and NGNB.
- 2) I agree with the reviewers that the slash marks between percentages make for difficult reading. I would remove them and use the format applied in the discussion.
- 3) What is the difference between "no results" and "absent" results?
- 4) There are still many typos and grammatical mistakes. A few are highlighted: p.4 lines 69-71, please rewrite for clarity; p.4 line 73, should it read "...but rarely cause disease"?; p. 5 line 87 "high" number. Please carefully do a manual read of the text.
- 5) p. 10, line 202, shouldn't this paragraph refer to Table 3 and not Table 2?
- 6) p. 7, line 137, I think it should read " A total of 14,478" not 11,478. Please double check all numbers in the text and tables.

Once these issues are addressed, I will forward your manuscript for acceptance and production.

Revision Guidelines

Sincerely,
Karen Carroll
Editor
Microbiology Spectrum

Sept. 25, 2024

Re: Spectrum02276-24 Multi-Year Comparison of VITEK MS performance for identification of rarely encountered pathogenic gram-negative bacilli (GNBs) in a large integrated Canadian health region

Response to the Reviewers:

Reviewer #1	
Comments:	Response to the Reviewers
1) Define the abbreviations in the Abstract. Use these abbreviations consistently throughout.	All abbreviations in the Abstract have been clarified; a consistent format for all abbreviations have been used throughout the paper.
2) Change slash marks between percentages	All uses of this format have been removed and a consistent format used throughout for clarity
3) Explain the difference between “no results” and “Absent”	Only “no results” have been used throughout. The term absent has been removed for clarity.
4) Fix typos and grammatical mistakes	The manuscript has been thoroughly edited and reviewed by a Spelling and Grammar software program.
5) Review citations for the Tables in the text	All Table citations have been reviewed. Line 202 has been corrected.
6) Double check all Tables and text	This has been done. All numbers in the Tables have been checked, corrected where necessary and changes made to the appropriate text.

Re: Spectrum02276-24R1 (Multi-Year Comparison of VITEK® MS performance for identification of rarely encountered pathogenic gram-negative bacilli (GNBs) in a large integrated Canadian healthcare region.)

Dear Dr. Deirdre L. Church:

Your manuscript has been accepted, and I am forwarding it to the ASM production staff for publication. Your paper will first be checked to make sure all elements meet the technical requirements. ASM staff will contact you if anything needs to be revised before copyediting and production can begin. Otherwise, you will be notified when your proofs are ready to be viewed.

Sincerely,
Karen Carroll
Editor
Microbiology Spectrum